# *Aspergillus flavus*-Mediated Green Synthesis of Silver Nanoparticles and Evaluation of Their Antibacterial, Anti-*Candida*, Acaricides, and Photocatalytic Activities

Amr Fouda [1,*], Mohamed A. Awad [2,*], Zarraq E. AL-Faifi [3], Mohammed E. Gad [2], Areej A. Al-Khalaf [4], Reham Yahya [5,6] and Mohammed F. Hamza [7,8]

1    Department of Botany and Microbiology, Faculty of Science, Al-Azhar University, Nasr City, Cairo 11884, Egypt
2    Department of Zoology and Entomology, Faculty of Science, Al-Azhar University, Nasr City, Cairo 11884, Egypt; mohamed.gad9@azhar.edu.eg
3    Center for Environmental Research and Studies, Jazan University, Jazan 42145, Saudi Arabia; zalfifi@jazanu.edu.sa
4    Biology Department, College of Science, Princess Nourah Bint Abdulrahman University, Riyadh 11671, Saudi Arabia; aaalkhalaf@pnu.edu.sa
5    Medical Microbiology, College of Science and Health Professions, King Saud Bin Abdulaziz University for Health Sciences, Riyadh 11671, Saudi Arabia; yahyar@ksau-hs.edu.sa
6    King Abduallah International Medical Research Center, Riyadh 11671, Saudi Arabia
7    School of Nuclear Science and Technology, University of South China, Hengyang 421001, China; m_fouda21@hotmail.com
8    Nuclear Materials Authority, El-Maadi, Cairo 11728, Egypt
*    Correspondence: amr_fh83@azhar.edu.eg (A.F.); mohamed_awad@azhar.edu.eg (M.A.A.); Tel.: +20-111-335-1244 (A.F.); +20-100-883-1007 (M.A.A.)

**Abstract:** *Aspergillus flavus* F5 was used to reduce $AgNO_3$ to form silver nanoparticles (Ag-NPs) that were monitored by a color change from colorless to yellowish-brown. The characterizations were achieved by UV-Vis spectroscopy, FT-IR, TEM, SEM-EDX, and XRD. Data showed that there was a successful formation of crystalline, spherical shape Ag-NPs with a particle average size of $12.5 \pm 5.1$ nm. The FT-IR clarified the role of various functional groups in the reducing/capping process. EDX-SEM revealed that the main component of the as-formed sample was set to be mainly Ag with a weight percentage of 46.1%. The synthesized Ag-NPs exhibit antibacterial and anti-*Candida* activity against *Staphylococcus aureus*, *Bacillus subtilis*, *Pseudomonas aeruginosa*, *Escherichia coli*, *Candida albicans*, *C. glabrata*, *C. tropicalis*, and *C. parapsilosis*, with inhibition zones ranging between $9.3 \pm 0.5$ to $20.8 \pm 0.3$ nm based on concentrations used and MIC values between 6.25 to 25 ppm. The mortality percentages of *Tyrophagus putrescentiae* mite species due to the mixing of their diet with different Ag-NPs concentrations of 0.5, 1.0, and 1.5 mg were $55.7 \pm 2.1$, $73.3 \pm 1.5$, and $87.4 \pm 1.6\%$ respectively after 20 days post-treatment. The catalytic activity of Ag-NPs to degrade methylene blue (MB) was investigated in the presence and absence of light irradiation. Data showed that a high photocatalytic degradation of MB compared with dark conditions at various times and concentrations. At a concentration of 70 mg/30 mL after 200 min., the dye removal percentages were $86.4 \pm 0.4\%$ in the presence of light irradiation versus $66.5 \pm 1.1\%$ in dark conditions.

**Keywords:** green synthesis; methylene blue; photocatalysts; *Tyrophagus putrescentiae*; antimicrobial

## 1. Introduction

Nanomaterials are defined as those materials that have surface structure or external dimensions at the nanoscale. The ISO (International Organization for Standardization) defined the nanoscale term in the range of 1–100 nm [1]. Recently, nanomaterials have been incorporated into everyday consumer goods due to their unique properties, such as optical,

electrical, mechanical, compatibility, stability, shape and size, charge, and high surface area, when compared with bulk counterparts [2–4]. Nanomaterials are synthesized by two major approaches: traditional (chemical and physical approaches) and green approaches. The traditional approaches have advantages such as producing different nanoparticles (NPs) with specific shapes and sizes, extensive scalability, and the vast applications of NPs produced by these approaches, such as electrical applications, energy conservation, catalytic activity, and disease therapy [5,6]. However, there are undeniable disadvantages or negative effects due to using these approaches, resulting in a decrease in their usage. These disadvantages include the reproductive and neurobehavioral effects of organic solvents involved in NPs' synthesis, harsh conditions such as heat, pressure, and pH used during different synthesis steps, and the production of volatile vapor and $CO_2$ which have remarkably negative impacts on both the environment and humans [7,8].

Green synthesis or biosynthesis is a low-cost method that is safe, clean, biocompatible, and environmentally friendly. With this method, biological entities such as microorganisms (bacteria, actinomycetes, yeast, fungi, and algae) and plant extract are utilized to reduce/cap and stabilize metal or metal oxide precursors to form NPs [9,10]. The reduction and capping steps are achieved by active metabolites secreted by a microorganism or those that exist in plant extract [11]. Fungi are an excellent source for the secretion of a variety of metabolites such as enzymes, proteins, and amino acids that hasten the green synthesis of NPs and impart high NPs' stability. In addition, fungi are easy to handle, have good heavy metal accumulators, easy scale-up production, high biomass production, low toxicity, and can synthesize NPs both intra- and extracellularly [12,13]. Different metal and metal oxides' NPs were synthesized, including Au, Ag, CuO, ZnO, and MgO, through harnessing fungal strains as environmental friendliness producers [7,13–15]. Therefore, in the current study, we hypothesize the high efficacy of the fungal strain *Aspergillus flavus* to be used as a biocatalyst for the green synthesis of one of the most promising NPs, silver nanoparticles (Ag-NPs).

Silver has been the metal most identified from ancient times as microbicidal [16]. Ag-NPs are one of the most common metal NPs to attract researchers' attention due to their new characteristics at the nanoscale. Among these characteristics are high thermal and electrical conductivity, high stability, catalytic activity, surface-enhanced Raman scattering, and optical properties [17]. The global production of Ag-NPs for industrial applications has significantly increased and is estimated to be 800 tons by 2025 [18]. Besides the broad spectrum of Ag-NPs against various pathogenic microbes, they can be used in fields such as cosmetics, textile, wound dressing, food packaging, biomedical devices, disinfectant products, agricultural sectors, wastewater treatment, preservation of historical papers, sensors, and batteries [13,19]. The global damage to stored grains by various mites and insect species is estimated at 5–10% in developed countries and 35% in developing countries [20].

Stored product pests (mites and insect species) are considered to be one of the biggest causes of food damage worldwide. Besides damaging grains, they can alter the food taste, cause human allergies, and be considered a vector for various pathogens [20]. Among stored-product mites, *Tyrophagus putrescentiae* feeds on the high fat and protein-content substances such as nuts, grain germ, seeds of oil rape, sunflowers, ham, cheese, and dog food [21]. The control of *T. putrescentiae* is especially difficult in dry surroundings because of its ecological, morphological, and physiological characteristics [22]. Therefore, it is urgent that new active compounds used in the biocontrol of mites are developed to overcome the resistant strains that have appeared due to the uncontrollable usage of chemical substances.

The main problem in industries such as textiles, chemicals, plastics, and cosmetics is the production of a large amount of contaminated water by dyes [23]. The discharge of this effluent without treatment causes serious concern to humans and the surrounding environment [24]. In most cases, traditional methods (chemical and physical methods such as filtration, adsorption, oxidation, and electrocoagulation) are used, but they are expensive and insufficient for most pollutants [25]. Moreover, the presence of high contaminants and non-biodegradable compounds can adversely affect microbial sludge and hinder

biological treatment [26]. The photocatalytic degradation of wastewater using nanoparticles is superior to traditional and biological methods because of its rapid oxidation, simplicity, low cost, efficiency, and low production of toxic by-products [27].

This study aims to discuss the green synthesis of Ag-NPs using fungal stains *A. flavus* and the characterization of the final product by UV-Vis spectroscopy, Fourier transform infrared (FT-IR), transmission electron microscopy (TEM), scanning electron microscopy connected with energy-dispersive X-ray (SEM-EDX), and X-ray diffraction (XRD). Further, the antibacterial activity against pathogenic Gram-positive and Gram-negative bacteria, anti-*Candida* against a different variety of coded and clinical *Candida* species, acaricides activity against *T. putrescentiae* mite, and photocatalytic degradation of methylene blue as dye model will be investigated.

## 2. Results and Discussion

### 2.1. Biosynthesis of Ag-NPs by A. flavus

Silver nanoparticles have received more attention from scholars due to their promising activities in several fields [28]. Therefore, attempts have increased to synthesize Ag-NPs using different methods. In the current study, Ag-NPs were synthesized by fast, simple, eco-friendly, and cost-effective methods (green method) by harnessing the metabolites of fungal strains to reduce/cap and stabilize $Ag^+$ to form $Ag^0$. The synthesis of metal and metal oxide NPs by green methods (using plant, fungi, bacteria, actinomycetes, and yeasts) is preferred over chemical and physical methods to avoid the disadvantages resulting from these methods [12]. Herein, the cell-free filtrate (CFF) of *A. flavus* strain F5 containing various metabolites was used as a biocatalyst for reducing $Ag^+$ to form Ag-NPs. *Aspergillus* spp. are the fungal strains most useful in nanobiotechnology due to their variety of organic acids and protein secretion, hence producing NPs with varied shapes and sizes [29]. In the current study, due to the unusual habitat of *A. flavus* used for Ag-NPs' synthesis (isolated from deteriorated historical paper), we predict their high efficacy in the secretion of a wide range of metabolites that impart high stability of synthesized Ag-NPs. The color change of CFF, from colorless to yellowish-brown, after mixing with $AgNO_3$ indicates the formation of Ag-NPs. This color change can be attributed to the excitement in the surface plasmon resonance of NPs [30,31]. The color intensity is related to the electrons produced from the reduction of $NO_3$ to $NO_2$ that are responsible for reducing $Ag^+$ to metallic ions ($Ag^0$) [32].

### 2.2. Characterizations of Ag-NPs

#### 2.2.1. UV-Vis Spectroscopy

The color change from colorless to yellowish-brown indicates the formation of Ag-NPs. The intensity of the formed color was checked by measuring its absorbance in the range of 300–700 nm to detect the maximum surface plasmon resonance (SPR). The maximum SPR for Ag-NPs fabricated by *A. flavus* F5 was observed at 415 nm after 24 h (Figure 1). After 15 days, the absorption was increased while no change in the maximum SPR peak was observed at 415 nm; this was attributed to the complete reduction of $Ag^+$ ions that strengthened the color intensity [31,33]. Udayasoorian et al. reported that the width and frequency of SPR depend on various factors such as the shape and size of metal NPs, surrounding media, and the metal-dielectric constant [34]. The maximum SPR peak localized at a wavelength between 410 to 420 is common for spherical shapes [35]. Moreover, various published studies reported that the observed SPR peak in the range of 400 to 460 nm indicates a successful synthesis of Ag-NPs [36]. Compatible with the obtained data, the SPR for Ag-NPs fabricated by different fungal strains such as *Penicillium italicum*, *Aspergillus sydowii*, *Penicillium chrysogenum*, and *Aspergillus niger* have SPR peaks in the ranges of 410–420 nm [33,35,37,38].

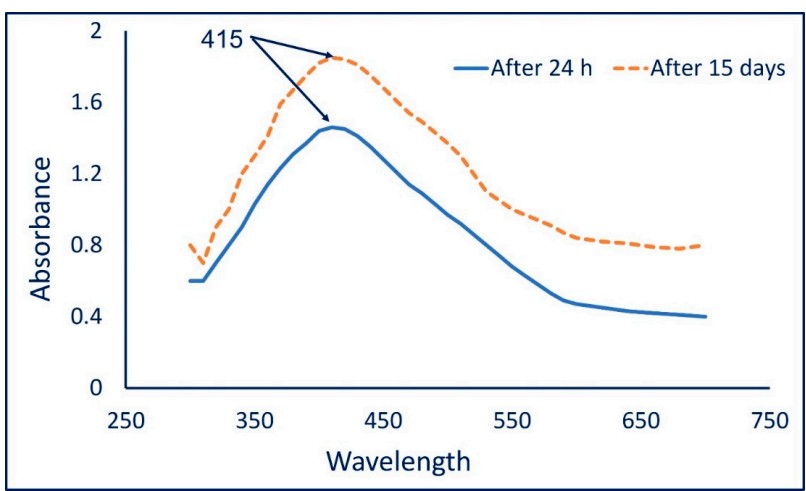

**Figure 1.** UV-Vis spectroscopy of biosynthesized Ag-NPs at different times (after 24 h and 15 days) referred to no change in SPR peak.

### 2.2.2. Fourier Transform Infrared (FT-IR)

The identification of various functional groups in the cell-free filtrate and their roles in capping and stabilizing synthesized Ag-NPs were analyzed by FT-IR. As shown in Figure 2, the FT-IR chart contains eight peaks at wavenumbers 595, 840, 1105, 1380, 1640, 2170, 2920, and 3450 $cm^{-1}$. The peak at 595 $cm^{-1}$ signifies an O–H out-of-plane (bend) of the produced polysaccharide moiety [39]. The weak peak at 840 $cm^{-1}$ is related to nitrate ions from the silver salt and C–O out-of-plane (bend) [40], whereas the broad peak at 1105 $cm^{-1}$ corresponds to C–O–C asymm. str., as well as the C–O stretching and $NH_2$ rock of the polysaccharide groups [41]. The strong peak at 1380 $cm^{-1}$ is signified by the C–N stretching of aliphatic and aromatic amines [42]. On the other hand, the strong peak at 1640 $cm^{-1}$ corresponding to the C=O and C=N (amide) stretching overlapped with N–H bending amines [43,44]. The peak observed at a wavenumber of 2170 $cm^{-1}$ is signified by isocyanate moiety (formed in the microorganism cell) –N=C=O [42]. This was supported by the data reporting the presence of C=O and C=N at 1640 $cm^{-1}$ [45]. Finally, the observed peak at 2920 $cm^{-1}$ corresponds to the vibration stretching of C–H of alkanes [46], while the broad peak at 3450 $cm^{-1}$ is correlated to the convolution of stretching N–H and O–H [47]. The obtained data confirmed the presence of several functional groups such as alkanes, alkenes, aliphatic and aromatic amines, and alkyl that exist in the cell-free filtrate of *A. flavus* and play an important role in the reduction, capping, and stabilizing of Ag-NPs.

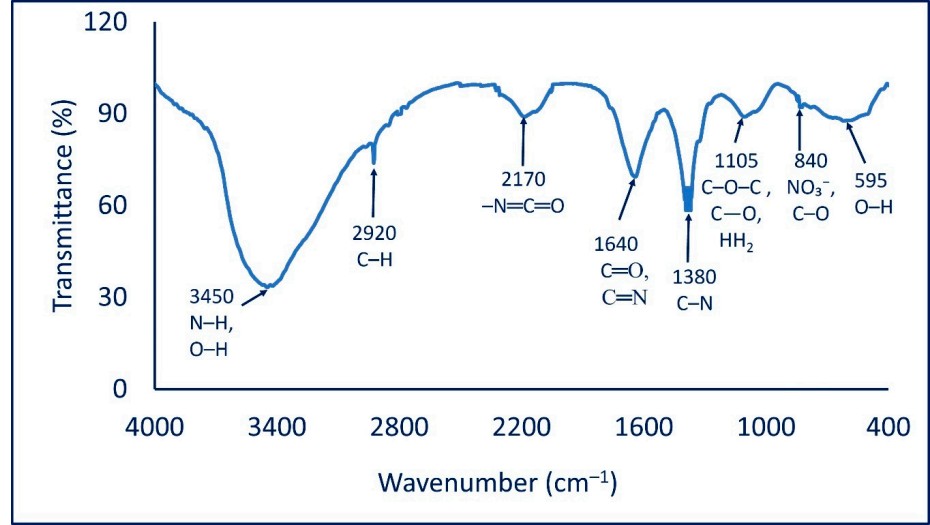

**Figure 2.** FT-IR chart of Ag-NPs fabricated by cell-free filtrate of *A. flavus*.

### 2.2.3. Transmission Electron Microscopy (TEM)

Transmission electron microscopy is a useful technique used for investigating the shape, size, aggregation, and other morphological characteristics of synthesized NPs. As shown, the metabolites secreted by fungal strain *A. flavus* F5 have the potential to reduce/cap AgNO$_3$ and form well-dispersed Ag-NPs with spherical shapes (Figure 3A). The size of biosynthesized Ag-NPs ranged from 3–28 nm with an average diameter of 12.5 ± 5.1 nm (Figure 3B). Compatible with the present study, spherical Ag-NPs were successfully formed by white-rot fungi *Bjerkandera* sp. R1 with a size ranging from 10–30 nm [14]. The activity of NPs is influenced by a variety of properties such as surface characteristics, size, coating or capping agent, shape, reactivity, and solubility [48]. It is worth noting that the toxicity of NPs differs based on their size and shape which affects the contact between NPs and cells, with toxicity increasing as size decreases [23]. For instance, the toxicity of Ag-NPs that have a size of 60 nm and 100 nm against mice was lower than those that have a particle size of 10 nm [49]. Moreover, Raza et al. successfully formed two shapes of Ag-NPs with varied sizes: spherical with sizes of 15 nm and 90 nm, and a triangular shape with a size of 150 nm. Raza et al. reported that the toxicity of spherical shapes with a size of 15 nm against two Gram-negative bacteria, *Escherichia coli*, and *Pseudomonas aeruginosa*, was greater than those reported by the larger size of spherical and triangular shapes [50]. Based on these data, we predict the high activity of synthesized Ag-NPs in the current study due to their smaller sizes.

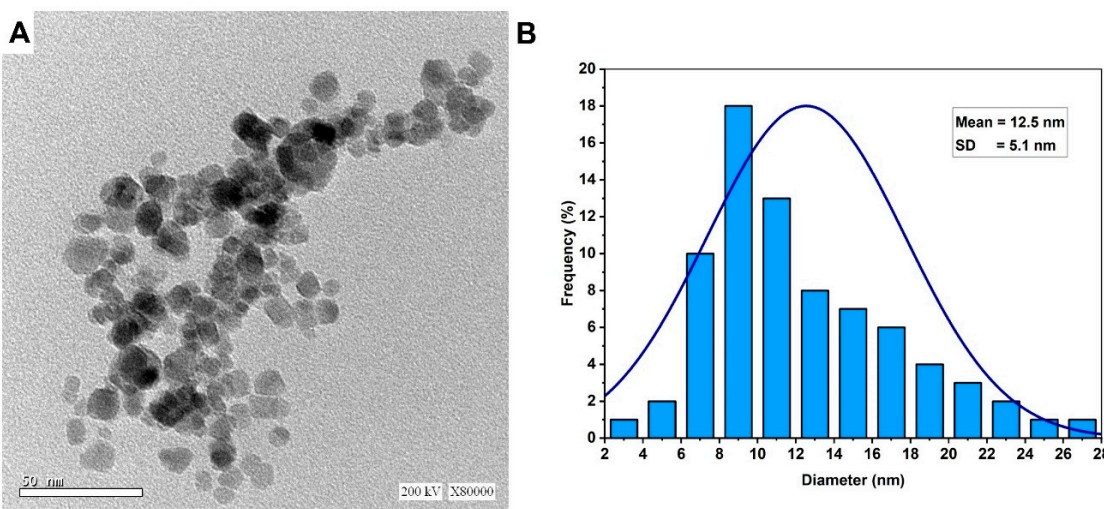

**Figure 3.** Characterization of Ag-NPs synthesized by *A. flavus* strain F5. (**A**) is the TEM image that showed a spherical shape and (**B**) is the size distribution according to the TEM image.

### 2.2.4. SEM-EDX

The surface morphology, qualitative, and quantitative elemental compositions of Ag-NPs synthesized by fungal strain were investigated by SEM connected with EDX apparatus. As shown, the synthesized Ag-NPs were well dispersed with a spherical shape (Figure 4A). The EDX chart exhibits a strong peak in the region of silver which indicates the successful formation of Ag-NPs (Figure 4B). Herein, the optical absorption peak of metallic Ag-NPs showed at 3 Kev because of their SPR, as previously reported [51,52]. Based on the EDX chart, the major component of the synthesized materials is Ag. As shown, C, O, and Ag were present with weight percentages of 35.5, 18.4, and 46.1%, respectively (Figure 4B). The presence of C and O can be attributed to the scattering of the capping agent by X-ray [53]. Amargeetha and Velavan [51] reported that the successful reduction of Ag$^+$ to Ag-NPs can be confirmed by EDX analysis by the strong thickness of the silver peak compared to other peaks in the chart. In the current study, the peak of Ag in the EDX chart was denser and thicker compared with other peaks of C and O, confirming the capacity of metabolites secreted by *A. flavus* to form Ag-NPs.

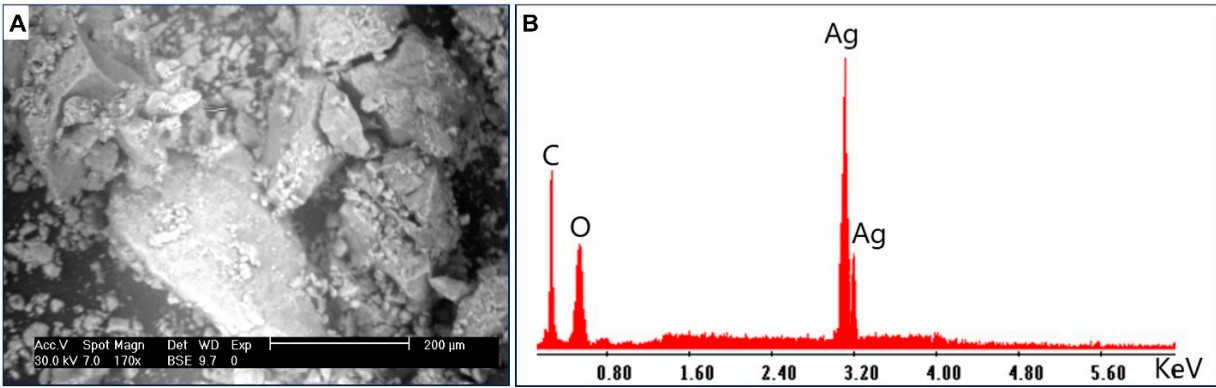

**Figure 4.** (**A**) is SEM analysis and (**B**) is the EDX chart of fungal-mediated biosynthesis of Ag-NPs.

### 2.2.5. X-ray Diffraction (XRD)

The crystallinity of biosynthesized Ag-NPs was investigated by XRD in the ranges of two theta values of 3–83 (Figure 5). The spectra reveal the presence of four Bragg reflection intense peaks at (111), (200), (220), and (311) at two theta values of 38.5°, 44.8°, 64.4°, and 77.5°, respectively. The presence of unassigned diffraction peaks could be attributed to the crystallization of biomolecules that exist in CFF and which are responsible for capping and stabilizing the synthesized NPs on its surface [51,54]. The obtained data showed the crystallinity face center cubic of *A. flavus*-mediated green synthesis Ag-NPs according to the standard of JCPDS card number 04-0783 [51]. Recently, the XRD spectra of *A. sydowii*-mediated green synthesis of Ag-NPs showed maximum peaks at two theta values of 38.3°, 44.5°, 64.6°, and 77.7° for (111), (200), (220), and (311), respectively [37]. The Debye-Scherrer equation was used to measure the particle size of synthesized NPs. Herein, the average Ag-NPs' size compatible with TEM analysis was 25 nm. The broadening of the bases of Bragg's diffraction peaks indicates the successful fabrication of small Ag-NPs sized as reported previously [55,56].

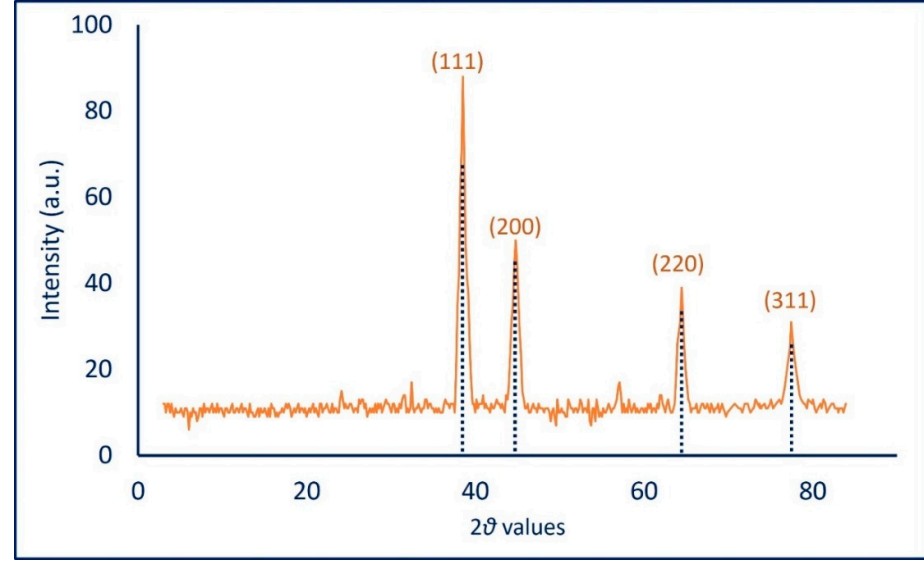

**Figure 5.** X-ray diffraction (XRD) analysis of Ag-NPs fabricated by *A. flavus*.

### 2.3. Antimicrobial Activity

The resistance of pathogenic microbes to antibiotics is considered the main challenge in medical sectors, retarding the successful diagnosis and hence treatment of diseases. Therefore, discovering new active compounds to reduce mortality and morbidity, especially those caused by multidrug-resistant microbes, is a matter of urgency [57]. In the current study, the activity of Ag-NPs synthesized by the green approach was investigated

against various coded and clinical pathogenic microbes represented as *S. aureus*, *B. subtilis*, *E. coli*, *P. aeruginosa*, *C. albicans*, *C. glabrata*, *C. tropicalis*, and *C. parapsilosis*. An analysis of variance showed that the activity of biosynthesized Ag-NPs was dose-dependent. The highest inhibition zones were recorded for Ag-NPs at a concentration of 100 ppm with a diameter of $18.7 \pm 0.5$, $17.7 \pm 0.6$, $20.7 \pm 0.6$, and $20.8 \pm 0.3$ mm for *B. subtilis*, *S. aureus*, *P. aeruginosa*, and *E. coli*, respectively (Figure 6A). The same result was achieved at the highest concentration against *Candida* spp., which recorded zones of inhibitions of $16.8 \pm 0.3$, $17.8 \pm 0.2$, $15.8 \pm 0.3$, and $15.4 \pm 0.3$ mm for *C. albicans*, *C. glabrata*, *C. tropicalis*, and *C. parapsilosis*, respectively (Figure 6B). Similarly, Ag-NPs synthesized by aqueous extract of *Parkia speciosa* showed promising antibacterial activity against *B. subtilis*, *S. aureus*, *P. aeruginosa*, and *E. coli* at 100 µg mL$^{-1}$ [58]. In addition, Ag-NPs fabricated by *Penicillium chrysogenum* showed high activity against various clinical *Candida* species represented by *C. albicans*, *C. tropicalis*, *C. krusei*, *C. parapsilosis*, and *C. glabrata* [38]. Taha et al. reported that the highest clear zones due to the treatment of *S. aureus*, *E. coli*, *C. albicans*, and *C. tropicalis* with Ag-NPs formed by *Penicillium italicum* were attained at 100 ppm with values of $17.6 \pm 2.2$, $19.5 \pm 0.5$, $35.0 \pm 0.5$, and $35.6 \pm 0.6$ mm, respectively [35].

In the current study, the activity of Ag-NPs decreased by decreasing their concentrations. For example, at 50 ppm, the formed inhibition zones were decreased to $15.3 \pm 0.5$, $14.7 \pm 0.6$, $18.5 \pm 0.5$, $18.8 \pm 0.3$, $14.7 \pm 0.6$, $15.3 \pm 1.2$, $13.3 \pm 0.6$, and $13.8 \pm 0.3$ mm for *B. subtilis*, *S. aureus*, *P. aeruginosa*, *E. coli*, *C. albicans*, *C. glabrata*, *C. tropicalis*, and *C. parapsilosis*, respectively. The detection of minimum inhibitory concentrations (MIC) for synthesized NPs against pathogenic microbes is the main target to possibly integrate into medication [59]. Herein, the MIC values of synthesized Ag-NPs were 12.5 ppm for *B. subtilis*, *C. albicans*, *C. glabrata*, *C. tropicalis*, and *C. parapsilosis*, 6.25 ppm for *P. aeruginosa* and *E. coli*, and 25 ppm for *S. aureus*, with inhibition zones ranging between $9.3 \pm 0.5$–$11.3 \pm 1.2$ mm. In our recent study, the MIC values of Ag-NPs synthesized by endophytic *Streptomyces antimycoticus* strain L-1 were 12.5, 25, 12.5, 25, and 12.5 ppm for *B. subtilis*, *S. aureus*, *E. coli*, *Salmonella typhi*, and *P. aeruginosa*, respectively, with clear zones ranging between $9.5 \pm 0.4$–$13.3 \pm 0.6$ mm [31].

The inhibitory mechanisms of Ag-NPs are varied and can be attributed to their efficacy to release silver ions that destroy the microbial cells [16]. The released Ag ions can attach to the cell wall and cell membrane, ultimately destroying the cell envelope by enhancing the cell membrane permeability [60]. Moreover, due to silver ions' adhesion, they can enhance the microbial cells and lead to the deactivation of the respiratory enzymes, enhance the production of reactive oxygen species (ROS), interrupt the production of ATP, react with phosphorus and sulfur in DNA to block their replication and then stop cell division, and cause ribosome denaturation that inhibits the protein synthesis [12].

Another inhibitory mechanism is an accumulation of Ag-NPs in the pits on the cell wall that follow their penetration into the cell membrane, which ultimately modifies or destroys the cell wall and cell membrane structure leading to the rupture of the cell components [61]. In addition, Ag-NPs can hinder the transduction of the bacterial signals through dephosphorylation of tyrosine residue that exists on the peptide substrate, ultimately leading to cell apoptosis and inhibiting cell reproduction [62].

In the current study, the prokaryotes (bacterial strains) are more susceptible to Ag-NPs than eukaryotes (*Candida* strains). Among prokaryotes, Gram-negative bacteria are more sensitive toward Ag-NPs than Gram-positive bacteria and this was attributed to the cell wall structure. The cell wall of Gram-positive bacteria is thicker than Gram-negative bacteria due to the presence of a high peptidoglycan layer. The thicker Gram-positive bacterial cell wall can decrease the penetration of Ag-NPs into the bacterial cell and hence reduce their inhibitory effects [63]. The inhibitory effect of active compounds against *Candida* species can be due to their activity to change the profile of sterol in the *Candida* cell wall by inhibiting the pathway of ergosterol biosynthesis [64,65]. Ergosterol is the major sterols content in yeast, it is synthesized in the endoplasmic reticulum, and it is responsible for rigidity, integrity, homogeneity, and fluidity of the plasma membrane [65].

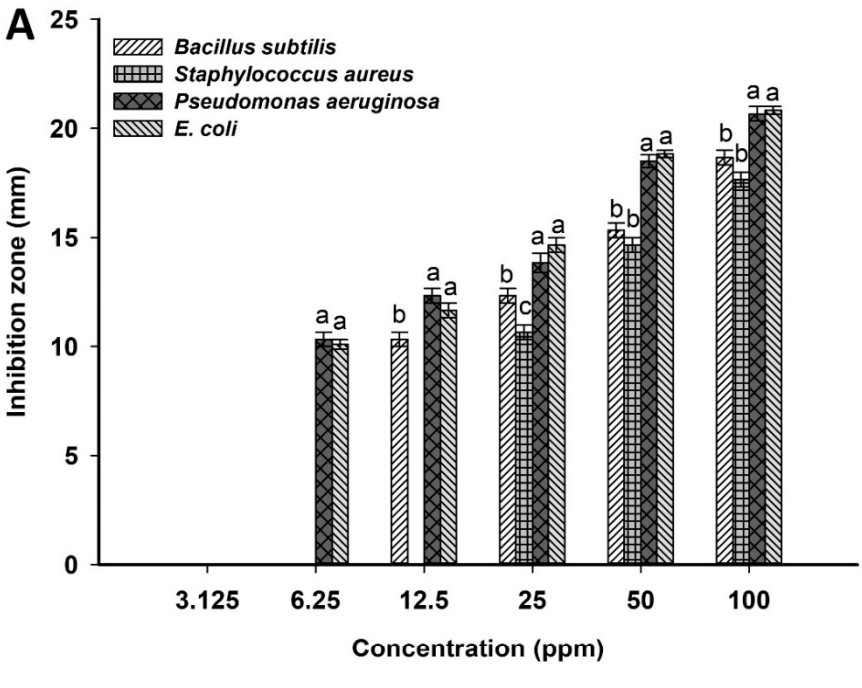

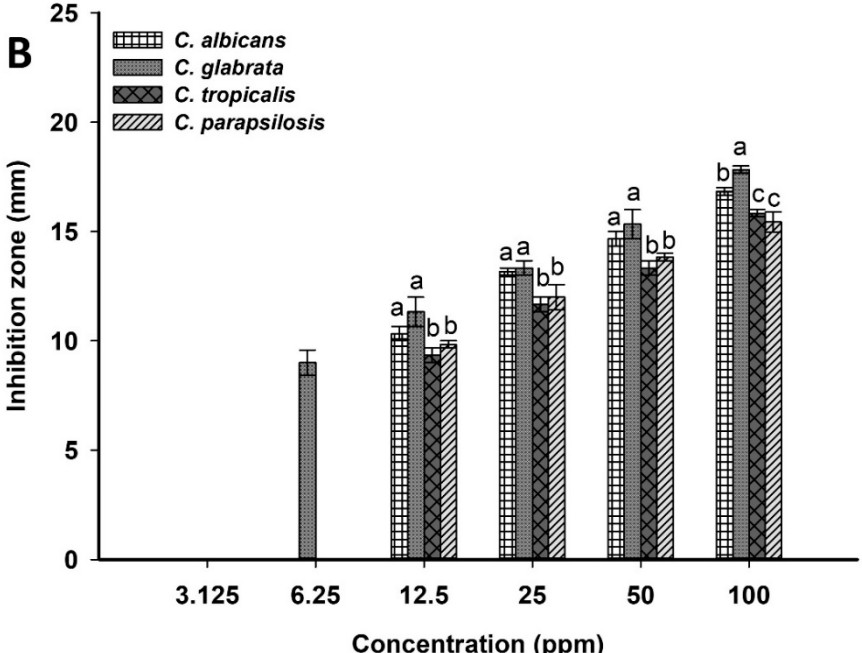

**Figure 6.** Antimicrobial activity of Ag-NPs synthesized by harnessing the metabolites of *A. flavus*. (**A**) is antibacterial activity against *B. subtilis*, *S. aureus*, *P. aeruginosa*, and *E. coli*, (**B**) is anti-candida activity against *C. albicans*, *C. glabrata*, *C. tropicalis*, and *C. parapsilosis*. Data are statistically different at $p \leq 0.05$ by Tukey's test, ($n = 3$); error bars are means $\pm$ SD. Different letters at the same concentration indicate the data are significant.

### 2.4. Acaricides Activity

The main strategy for the control of mites associated with various grains such as wheat and rice depend on the use of synthetic chemical acaricides. Although it seems to be a successful strategy, the abuse of these materials produces more resistance strains besides their negative impacts on human health, the environment, and beneficial microflora [66,67]. Therefore, the discovery of new compounds that are active, eco-friendly, reduced flammability, and low-cost is crucial. Nanomaterials, especially those synthesized by green methods, provide these advantages [68]. Tyagi et al. reported that the biosynthesized Ag-NPs and

gold nanoparticles (Au-NPs) were more effective and had the lowest toxicity compared with those synthesized by chemical methods [69]. *Tyrophagus putrescentiae* is the most common mite in stored grains, soil, house dust, and causing allergies in humans. It harbors various bacterial communities and can be considered a source for microbial spreading in soil and storage containers via their fecal pellets [70]. Therefore, it is urgent to control this mite using the green approach. To the best of our knowledge, this is the first report about the use of green synthesized Ag-NPs in the control of *T. putrescentiae*. Data presented in Table 1 showed that the activity of Ag-NPs against *T. putrescentiae* was time- and concentration-dependent. Data analysis showed that the mortality percentages were $42.3 \pm 1.5$, $48.3 \pm 2.3$, and $55.7 \pm 2.1\%$ for 0.5 mg Ag-NPs after 10, 15, and 20 days, respectively. These mortality percentages were increased by increasing the NPs concentrations to $73.3 \pm 3.8$, $79.7 \pm 2.5$, and $87.4 \pm 1.6\%$ for 1.5 mg after 10, 15, and 20 days, respectively. Data recorded by Jalal-izand et al., which reported that the mortality percentages of *Tetranychus urticae* mite due to treatment with Ag-NPs were concentration- and time-dependent, are compatible with our study. The highest mortality was recorded for Ag-NPs concentrations of 1000, 2000, and 3000 ppm after 144 h [71]. Moreover, the green synthesized Ag-NPs by *Beauveria bassiana* and *Metarhizium brunneum* was more highly effective against *Tetranychus urticae* than commercial chemical synthesized Ag-NPs and the precursor material of AgNO$_3$ [72]. The Ag-NPs synthesized by the chemical method exhibited high toxicity against *Tetranychus urticae* with mortality percentages of 85.21%, whereas they showed only a moderate effect against *Euseius scutalis* and *Neosiulus cucumeris* with mortality percentages of 23.77% and 9.95% at the highest concentration after one week [73].

**Table 1.** The mortality percentages of green synthesized Ag-NPs against *T. putrescentiae* at different concentrations after interval times.

| Ag-NPs Concentration | Mortality Percentages (%) * | | |
|---|---|---|---|
| | **10 Days** | **15 Days** | **20 Days** |
| 0.5 mg | $42.3 \pm 1.5$ [a] | $48.3 \pm 2.3$ [b] | $55.7 \pm 2.1$ [c] |
| 1.0 mg | $57.7 \pm 1.6$ [a] | $68.6 \pm 2.5$ [b] | $73.3 \pm 1.5$ [c] |
| 1.5 mg | $73.3 \pm 3.8$ [a] | $79.7 \pm 2.5$ [b] | $87.4 \pm 1.6$ [c] |

* Mortality was expressed as the mean $\pm$ SD (standard deviation) of three replicates ($p \leq 0.05$). Different letters in the same row indicate the data are significant.

An analysis of variance showed that there are significant differences between times for pre-oviposition, oviposition, and post-oviposition of adult females due to different Ag-NPs concentrations mixed with mite diets (wheat) compared with the control (wheat in the absence of Ag-NPs). Data showed that the pre-oviposition, oviposition, and post-oviposition periods of adult females were ($1.45 \pm 0.31$, $1.50 \pm 0.28$, and $1.59 \pm 0.18$ days), ($5.06 \pm 0.68$, $5.01 \pm 0.47$, and $4.55 \pm 0.43$ days), and ($1.38 \pm 0.24$, $1.44 \pm 0.29$, and $1.36 \pm 0.32$ days) due to treatment with 0.5, 1.0, and 1.5 mg, respectively, compared with the control (Table 2). Besides the efficacy of Ag-NPs on mortality percentages and oviposition periods, the fertility of mites via the capability of egg-laying was investigated. Interestingly, the fecundity (maximum output of the eggs by adult females) was high decreased due to feeding on Ag-NPs. Data showed that the total average of output eggs were $162.34 \pm 6.25$, $115.31 \pm 10.4$, and $54.81 \pm 6.1$ eggs/day after treatment with Ag-NPs at a concentration of 0.5, 1.0, and 1.5 mg, respectively, compared with the control ($325.40 \pm 12.1$ eggs/day) (Table 2). Therefore, it can be concluded that the high concentration of green synthesized Ag-NPs has a sterile effect on the *T. putrescentiae* female due to their effect on the growth of the ovary. Also compatible with our study, the capability of *Drosophila* to lay their eggs is decreased after treatment with different concentrations of Ag-NPs (5, 25, 50, and 250 mg/L) [3].

**Table 2.** Duration times of adult females of *T. putrescentiae* fed on wheat mixed with different Ag-NPs concentrations during pre-oviposition, oviposition, and post-oviposition periods, and the fecundity at $25 \pm 2$ °C.

| Treatment of Adult Females with Different Concentrations of Ag-NPs | Oviposition Period (Days) | | | Fecundity (Total Average of Eggs/Day) |
|---|---|---|---|---|
| | Pre-Oviposition | Oviposition | Post-Oviposition | |
| Control (Non-treated) | $1.33 \pm 0.45$ | $6.35 \pm 0.5$ | $1.79 \pm 0.27$ | $325.4 \pm 12.1$ |
| 0.5 mg | $1.45 \pm 0.31$ | $5.06 \pm 0.68$ | $1.38 \pm 0.24$ | $162.34 \pm 6.25$ |
| 1.0 mg | $1.5 \pm 0.28$ | $5.01 \pm 0.47$ | $1.44 \pm 0.29$ | $115.31 \pm 10.4$ |
| 1.5 mg | $1.59 \pm 0.18$ | $4.55 \pm 0.43$ | $1.36 \pm 0.32$ | $54.81 \pm 6.1$ |

*2.5. Photocatalytic Activity*

The catalytic activity of green synthesized Ag-NPs was assessed by the degradation of MB as a dye model. The catalytic efficacy was achieved at different Ag-NPs concentrations for different contact times under dark and light irradiation. An analysis of variance revealed that the degradation efficacy either in dark or light irradiation was dose- and time-dependent. The obtained data are compatible with those obtained by Mavaei et al., who reported that the activity of Ag-NPs to degrade three different dyes designated as methylene blue, new fuchsine, and erythrosine B were time-dependent [74]. The surface area of nano-catalysts is considered the main factor determining their catalytic efficiency because the reactions are accomplished on the surface region [75]. Nano-catalysts are characterized by smaller sizes with high surface area. Therefore, catalytic activity will be increased by decreasing the NPs' sizes [76]. In the current study, due to the smaller Ag-NPs' size, we can predict their high catalytic activity. At low Ag-NPs concentration (30 mg/30 mL) in dark conditions, the decolorization percentages were $32.01 \pm 3.3\%$ after 20 min and increased to $60.3 \pm 0.2\%$ and $60.5 \pm 0.5\%$ after 160 and 200 min, respectively, compared with the control's low decolorization percentages ($5.5 \pm 3.2\%$ after 20 min and $9.3 \pm 1.2\%$ after 200 min). However, under light irradiation, the decolorization percentages were $38.4 \pm 2.2\%$ after 20 min and $77.2 \pm 2.3\%$ and $80.7 \pm 0.8\%$ after 160 min and 200 min, respectively, at the same concentration (30 mg/30 mL) (Figure 7). By increasing Ag-NPs concentrations, the decolorization percentages were increased and this can be attributed to either increasing the number of adsorption sites or enhancing the hydroxyl radical's production by increasing NPs concentrations [77,78]. At high Ag-NPs concentrations (70 mg/30 mL), data analysis showed that there is no significant difference between decolorization percentages in dark and light irradiation after 20 min ($46.1 \pm 1.9\%$ in dark and $46.9 \pm 0.6\%$ in light conditions). By increasing time, the decolorization percentages were significantly different in dark and light conditions ($p \leq 0.001$). The decolorization percentages were $65.2 \pm 0.7\%$ in dark conditions and reached $82.9 \pm 0.9\%$ under light irradiation after 160 min (Figure 7). Data analysis showed that there is no significant difference in decolorization percentages of MB at times 160 and 200 min in dark conditions, whereas the significant difference was achieved in the presence of light. The degradation of dyes by nanomaterials depend on factors such as the type of dyes, type of adsorbent and their concentrations, and pH, as previously reported [79]. Nga et al. reported that the attraction between dyes and NPs can be attributed to the electrostatic force between negative and positive charges in dyes and NPs, respectively [80].

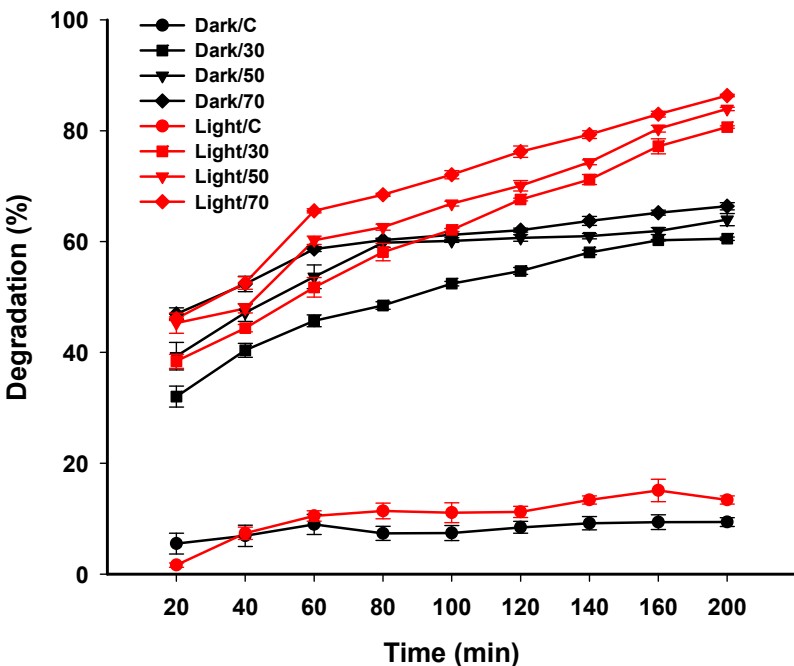

**Figure 7.** Photocatalytic activity of green synthesized Ag-NPs for degradation of methylene blue (MB) in dark and light irradiation. Data are statistically different at $p \leq 0.05$ ($n$ = 3); error bars are means $\pm$ SD. The standard deviation is less than the size of the symbols if no error bars are seen.

According to the obtained data, the decolorization of MB using Ag-NPs under light irradiation was better than those recorded in dark conditions. The researchers suggested three different mechanisms for dyes' color removal using nanomaterials. The first is a conversion of the dye to its white indigo (leuco form) in an alkaline medium or by reduction. The second mechanism is the adsorption of dye over the surface of nanomaterials due to the high surface area of NPs and the high number of dyes that can be removed by this mechanism. In the current study, the second mechanism looks to be efficient for the removal of MB by Ag-NPs in dark conditions [81]. The third mechanism is the degradation of dyes on the NPs' surface; because hot electrons are found on the surface formed by electron transition due to the SPR, this mechanism seems to be the most appropriate for the degradation of MB by Ag-NPs under light irradiations [82]. In the presence of light, electrons are excited from the valence band (VB) to the conducted band (CB) and hence form electron-hole pairs (Ag ($e^-_{CB}$ and $h^+_{VB}$)) (Figure 8). The $h^+_{VB}$ reacts with $H_2O$ forming hydroxyl radicals ($^\bullet OH$) and $H^+$, whereas the $e^-_{CB}$ reduces $O_2$ forming $^\bullet O_2^-$ (superoxide radicals) and $^\bullet OOH$ (hydrogen peroxide radicals). Finally, the various active radical species ($^\bullet OH$, $^\bullet O_2^-$, and $^\bullet OOH$) react with MB, leading to enhanced dyes degradation [74,83]. Jose et al. reported that the photocatalytic activity of nanomaterials can be attributed to photogenerated holes (that undergo oxidation conditions to form hydroxyl radicals) and electrons (that react with molecular oxygen to form oxygen anion radicals) causing a complete breakdown of dyes to $CO_2$, $H_2O$, and small ions [84]. In the current study, the synthesized Ag-NPs showed high antimicrobial and catalytic activity. Therefore, the benefit from this phenomenon is obtaining wastewater that is microbe and dye-free, and hence can be reused or discharged into the ecosystem in a safe mode.

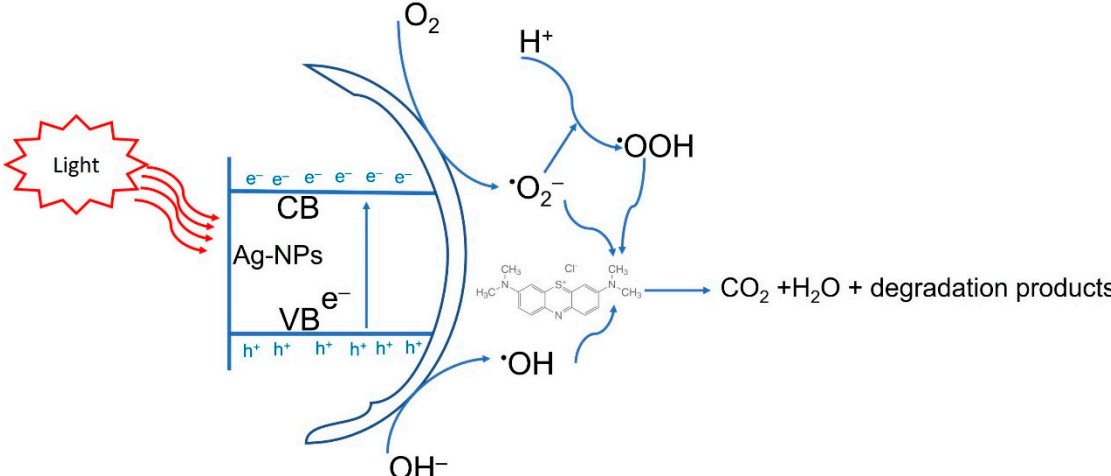

**Figure 8.** Prospective photocatalytic degradation mechanism of methylene blue using green synthesized Ag-NPs.

## 2.6. Reusability of Photocatalyst Ag-NPs

The photocatalyst stability or reusability is the critical factor for integration into industrial applications. In the current study, the reusability of green synthesized Ag-NPs in the photodegradation of MB was achieved under optimum conditions (in the presence of light irradiation at a concentration of 70 mg/30 mL for 200 min). The reusability test was measured for the fourth cycle. Data analysis showed that a 2.9% decrease in the activity of the catalyst was measured after four cycles. The photodegradation of MB was 86% after the first cycle and decreased to 83.1% after the fourth cycle (Figure 9). These modest reductions in photocatalyst activity can be related to the saturated sites that exist on the surface of catalysts with small intermediate biodegradable compounds [85]. Mavaei et al. reported that the photocatalytic activity of Ag-NPs during degradation of new fuchsine was decreased to 8% after the fourth run. The study reported that photodegradation was reduced from 98% after the first cycle to 88% after the fourth cycle [74]. Based on the obtained data, it can be concluded that the high stability of green synthesized Ag-NPs enhanced the photodegradation of MB for different cycles.

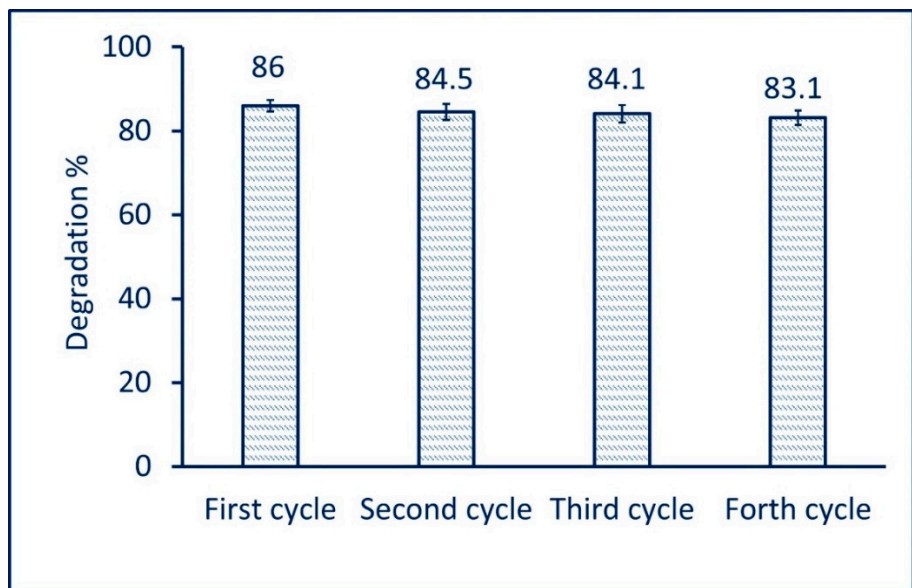

**Figure 9.** The reusability of green synthesized Ag-NPs for photodegradation of MB after four cycles.

## 3. Materials and Methods

### 3.1. Fungal Strain

In the current study, the silver nanoparticles were synthesized using the fungal strain *Aspergillus flavus* F5. This strain was isolated previously from a deteriorated historical paper collected from the Al-Azhar Library, Cairo, Egypt. The fungal strain was subjected to morphological characterization (cultural and microscopic examination) and molecular identification by amplification and sequencing of the ITS gene [86]. The identified gene sequence of strain F5 was deposited in GenBank with accession number MK452262.

### 3.2. A. flavus-Mediated Biosynthesis of Ag-NPs

The fungal strain F5 was inoculated into Czapek-Dox broth media and incubated at $30 \pm 2$ °C for five days at shaking conditions (120 rpm). The growth media then underwent filtration using Whatman filter paper (No. 1) to collect the fungal mass which was washed thrice with distilled water to remove adhering component media. The collected fungal biomass (10 g) was suspended in 100 mL of distilled $H_2O$ for 24 h under shaking conditions (120 rpm). At the end of the incubation period, the previous suspension was centrifuged at 10,000 rpm for 10 min to collect the supernatant (cell-free filtrate) which was then used for the biosynthesis of Ag-NPs as follows: 16.9 mg of metal precursor ($AgNO_3$) was dissolved in 10 mL distilled $H_2O$ and added to 90 mL of collected cell-free filtrate (CFF) to obtain a final concentration of 1 mM. The pH of the mixture was adjusted at 8 through dropwise of 1 M NaOH under stirring conditions. The formation of Ag-NPs was checked via a color change of the mixture from colorless to yellowish-brown [9]. The resultant was oven-dried at 100 °C for 24 h, then the residue was collected, washed with distilled $H_2O$ to remove any impurities, and left to dry before being used. The control (aqueous solution of $AgNO_3$ and cell-free filtrate in the absence of $AgNO_3$) was running with the experiment under the same conditions.

### 3.3. Characterizations

The maximum absorbance of biosynthesized Ag-NPs was detected using UV-Vis spectroscopy (JENWAY 6305, Cole-Parmer, Staffordshire, UK) by measuring the absorbance of aqueous solution at different wavelengths in the range of 300 to 700 nm. The Fourier transform infrared (FT-IR) analysis (Cary 630 FTIR model, Tokyo, Japan) was used to detect the different functional groups responsible for the reduction and stabilization of biosynthesized Ag-NPs. The FT-IR analysis was achieved by the KBr method as follows: 300 mg of biosynthesized Ag-NPs was well mixed with KBr under high pressure to form a disk that scanned at a wavenumber of 400 to 4000 cm$^{-1}$ [37]. The transmission electron microscopy (TEM JEOL 1010, Tokyo, Japan) was used to investigate the size and shape of biosynthesized Ag-NPs. Briefly, the TEM grid (carbon/copper) was loaded with a few drops of Ag-NPs solution and remained to complete adsorption, followed by touching with blotting paper to remove solution excess before being subjected to analysis. The composition of biosynthesized Ag-NPs was assessed by scanning electron microscopy connected with energy-dispersive X-ray (SEM-EDX) (JEOL, JSM-6360LA, Tokyo, Japan) [15]. The X-ray diffraction (XRD) (X′ Pert Pro, Philips, Eindhoven, the Netherlands) was used to investigate the crystallographic structure of biosynthesized Ag-NPs. The sample was analyzed at 2 theta values of 3 to 90. The XRD analysis was performed at a voltage and current of 40 KV and 30 mA, respectively, with Cu-Ka as a radiation source ($\lambda$ = 1.54 Å). The average size of Ag-NPs was calculated based on XRD analysis using the Debye-Scherrer equation as follows:

$$D = 0.9 \times \lambda / \beta \cos\theta \tag{1}$$

where D is the mean of average particle size, 0.9 is Scherrer's constant, $\lambda$ is the X-ray wavelength (1.54), and $\beta$ and $\theta$ are half of the maximum intensity and Bragg's angle, respectively.

### 3.4. Antimicrobial Activity

The activity of green synthesized Ag-NPs against various Gram-positive bacteria (*Staphylococcus aureus* ATCC6538 and *Bacillus subtilis* ATCC6633), Gram-negative bacteria (*Pseudomonas aeruginosa* ATCC9022 and *Escherichia coli* ATCC8739), and various *Candida* spp. (*C. albicans* ATCC10231, *C. glabrata*, *C. tropicalis*, and *C. parapsilosis*) were assessed by the agar well diffusion method [87]. The *Candida* spp. other than *C. albicans* were clinical strains, isolated, and identified in mycology Laboratory, National Research Centre, Cairo, Egypt, by Prof. M. M. Effat. Approximately 50 µL of overnight bacterial and *Candida* strains (growth adjusted at O.D. 1) were inoculated separately onto 100 mL of Mueller-Hinton agar and yeast extract peptone dextrose (YEPD) agar media respectively, shaking well, and poured into Petri dishes under aseptic conditions. After solidification, three wells (6 mm) were prepared in each inoculated plate before being filled with 100 µL of Ag-NPs solution (100 ppm). The loaded plates were refrigerated for 1 h before incubation at $35 \pm 2$ °C (for bacteria) and $30 \pm 2$ °C (for *Candida*) for 24 h. The results were recorded as the diameter of the clear zone (mm) formed around each well. The same steps were achieved with different concentrations of Ag-NPs (50, 25, 12.5, 6.25, and 3.125 ppm) to detect the minimum inhibitory concentration (MIC). The experiment was replicated in triplicate.

### 3.5. Acaricides Activity

3.5.1. Mite Cultures, Identifications, and Rearing

The culture of test mites *Tyrophagus putrescentiae* (Schrank) (Sarcoptiformes: Acaridae) was collected from wheat and rice grains (not subjected to any acaricides agents). Approximately 100 g of collected grains were incubated for 24 h under 40-W electric lamps. The appeared mites were transferred into Petri dishes by 0.3 mm camel hairbrush for further examination under a stereomicroscope. The identification of *T. putrescentiae* was achieved based on their morphometric characteristics [88]. The pure mite cultures were reared on plastic cups ($15 \times 12 \times 5$ cm) containing 40 g of diet (dried yeast: fry feed no. 1; 1:1 *w/w*) and incubated in dark conditions at $25 \pm 2$ °C, 75% RH (relative humidity) [89]. The fry feed was purchased from Korea Special Feed Meal, Chonju, South Korea. The experiment was conducted in the Animal House, Zoology Department, Faculty of Science, Cairo, Egypt.

3.5.2. Toxicity Assay

The toxicity of green synthesized Ag-NPs against the adult of *T. putrescentiae* was assessed at different times. Briefly, 20 mite adults were reared on wheat mixed with different concentrations of Ag-NPs (0.5, 1.0, and 1.5 mg/10 g wheat grains) and incubated at $25 \pm 2$ °C, 75% RH. The experiment was replicated in triplicate. Dead mites were detected daily and the mortality percentages were measured after 10, 15, and 20 days post-treatment using the following equation [90]

$$\text{Mortality percentages } (\%) = 1 - \frac{\text{n in T after treatment}}{\text{n in Co after treatment}} \times 100 \qquad (2)$$

where: n is the mite population, T = treated, and Co = Control.

Moreover, the oviposition periods for adult females and the fecundity (total number of eggs) were calculated.

### 3.6. Photocatalytic Activity

The catalytic activity of green synthesized Ag-NPs was investigated by degradation of methylene blue (MB) under dark and light irradiation. The experiment was achieved on 30 mL of MB solution mixed with different concentrations of Ag-NPs (30, 50, and 70 mg) for different contact times (0.0, 20, 40, 60, 80, 100, 120, 140, 160, and 200 min). The light source was a 250-Watt halogen lamp. Briefly, the MB solution was prepared at a concentration of 10 mg L$^{-1}$, followed by adding a specific Ag-NPs concentration to 30 mL of MB solution. The previous mixture was stirred for 30 min before the experiment to attain the absorption/desorption equilibrium. The mixture was then incubated at room

temperature under aeration and subjected to light irradiation. The same experiment was repeated under the same conditions in dark irradiation for a comparative study. After regular interval times, approximately 2.0 mL of each treatment was taken and subjected to centrifugation at 10,000 rpm for 5 min, with the optical density of clear supernatant measured (using spectrophotometers, M-ETCAL, OK International Ltd., Eastleigh, UK) at λ max of MB (664 nm). The percentages of MB color removal were calculated according to the following equation [83]:

$$D\% = \frac{MB_i - MB_I}{MB_i} \times 100 \tag{3}$$

where D % is the decolorization percentage, $MB_i$ is the initial absorbance, and $MB_I$ is the final absorbance.

The reusability of catalyst in MB degradation was achieved under optimum conditions for the fourth cycle. The catalyst was collected from the first cycle by centrifugation, washed twice with distilled $H_2O$, and subjected to oven-dry at 80 °C to remove water content before being used in the second cycle.

### 3.7. Statistical Analysis

All data presented in the current study are the means of three independent replicates. Data were analyzed using the statistical package SPSS v17. The mean difference comparison between the treatments was analyzed by t-test or the analysis of variance (ANOVA) and, subsequently, by Tukey's HSD test, at $p \leq 0.05$.

## 4. Conclusions

In the current study, Ag-NPs were successfully formed by reducing the metal precursor ($AgNO_3$) using metabolites secreted by fungal strain *A. flavus* F5 as a green approach. The synthesized NPs were characterized by a color change from colorless to yellowish-brown, UV-Vis spectroscopy, FT-IR, TEM, SEM-EDX, and XRD. Data verified the role of different metabolites such as alkanes, alkenes, aliphatic and aromatic amines, and alkyl in the fabrication process, as shown in FT-IR analysis. In addition, TEM and XRD analyses confirmed the spherical shape, crystalline structure, and particle sizes of 3–28 nm with an average diameter of 12.5 ± 5.1 nm. The composition of the as-formed sample was C, O, and Ag with weight percentages of 35.5, 18.4, and 46.1%, respectively, as shown in SEM-EDX analysis. The biological activities of synthesized Ag-NPs, including antibacterial, anti-Candida, acaricides, and photocatalytic activity, were investigated. Data showed a promising efficacy of green synthesized Ag-NPs to inhibit the growth of pathogenic prokaryotes (G+ and G− bacteria) and eukaryotes (*Candida* spp.). Moreover, the *T. putrescentiae* mite that causes agricultural grains damage can be controlled by Ag-NPs. Data showed that the highest mortality percentages of *T. putrescentiae* were 73.3 ± 3.8, 79.7 ± 2.5, and 87.4 ± 1.6% for 1.5 mg Ag-NPs mixed with diet after 10, 15, and 20 days, respectively. Surprisingly, green synthesized Ag-NPs showed high photocatalytic activity to degrade MB dye. Data analysis revealed that the highest catalytic degradation of MB was 86.4 ± 0.4% in the presence of 70 mg of catalyst and light irradiation after 200 min compared with dark conditions (66.4 ± 1.1%). Based on the obtained data, the promising activities for Ag-NPs synthesized by the green approach in various fields can be inferred.

**Author Contributions:** Conceptualization, A.F. and M.A.A.; methodology, A.F., M.A.A., M.E.G. and M.F.H.; software, A.F., M.E.G., Z.E.A.-F. and M.F.H.; validation, A.F., M.A.A., and M.E.G.; formal analysis, A.F., M.A.A., Z.E.A.-F., M.E.G. and M.F.H.; investigation, A.F., R.Y. and A.A.A.-K.; resources, A.F. and Z.E.A.-F.; data curation, A.F., M.A.A., Z.E.A.-F., M.E.G. and M.F.H.; writing—original draft preparation, A.F., M.A.A., Z.E.A.-F., R.Y., A.A.A.-K. and M.E.G.; writing—review and editing, A.F.; visualization, A.F., M.A.A., Z.E.A.-F., M.E.G., R.Y., A.A.A.-K. and M.F.H.; supervision, A.F. and M.A.A.; project administration, A.F.; funding acquisition, R.Y., A.A.A.-K. and Z.E.A.-F. All authors have read and agreed to the published version of the manuscript.

**Funding:** This research received no external funding.

**Data Availability Statement:** Data sharing is not applicable to this article.

**Acknowledgments:** The authors would like to thank Princess Nourah bint Abdulrahman University Researchers supporting project number (PNURSP2022R37), Princess Nourah bint Abdulrahman University, Riyadh, Saudi Arabia. Authors extend their appreciation to the Botany and Microbiology Department, Faculty of Science, Al-Azhar University, Cairo, Egypt, Zoology and Entomology Department, Faculty of Science, Al-Azhar University, Cairo, Egypt, and the Center for Environmental Research and Studies, Jazan University, Jazan, Saudi Arabia, for the great cooperation in the current study.

**Conflicts of Interest:** The authors declare no conflict of interest.

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
