# Peer review of "Aspergillus flavus-Mediated Green Synthesis of Silver Nanoparticles and Evaluation of Their Antibacterial, Anti-Candida, Acaricides, and Photocatalytic Activities"

_catalysts, doi:10.3390/catal12050462_

Round 1

Reviewer 1 Report

The paper presents a green synthesis of silver NPS combined with their reactivity test for bactericide against T. putrescentiae mite and photocatalytic degradation of bleu methylene as dye model. It is an interesting paper, however some revisions are required. They are listed below:

  • The authors disclaimed the traditional approaches for nanoparticles synthesis to introduce biosynthesis. This is probably exaggerated. Traditional approaches have some advantages. In addition the advantages of biosynthesis are more than just replace the traditional approaches.  The authors describe the advantages in the following paragraphs. So thay should be more moderate in the description of traditional approaches.
  • In the introduction, some transition sentences are missing between the paragraph to introduce the different ideas.
  • In the introduction, the authors should justify why the synthesis of silver NPs by the chosen fungal strain is important in comparison to the already published synthesis by other biosources.
  • It is not clear how the silver nanoparticles are extracted before IR analysis and use.
  • On the TEM, the particles are not well dispersed. Another image with better dispersion is needed in particular to determine the size distribution of the particles.
  • Please clarify the argument for Ag(0) determined by EDX (P6-line 201-203)
  • The size determined by EDX is relative to a crystal domain. The size is comparable to TEM measurement, so the particles are mainly monocrystalline. Do you have any other analysis to confirm this point ? (HRTEM)
  • Silver nanoparticles can be toxic, so a balance between toxicity versus efficacy against T. putrescentiae mite should be discussed. In particular for a non negligeable quantity (0.5 mg and 1.5mg!). In addition the advantages of this kind of particles in comparison to classical one (obtained by classical approaches) should be discussed.
  • The concentration written as 30mg/30ml is strange why don’t use 1mg/ml ?
  • For the photocatalysis; please revise the proposed mechanism to better explain what happen in the dark. The differences are not so high between ligth and dark. What promotes the electron hole separation in this system? What is the motor for activity in the dark.

Author Response

Dear reviewer thank you very much for your valuable comments. All comments were answered point by point. Please see the attachment.

Reviewer 2 Report

The work shown for evaluation deals with the obtaining in a green way of silver nanoparticles and their antibacterial and photocatalytic behavior. The title and last affiliations seems not to follow the article template (spacing); in Abstracts lines 32-34 should be removed or reconsidered after improvement, as well as lines 58-60 in Introduction. 87 references ends the manuscript.

-why is an increase in the absorbance intensity after 15 days (Figure 1)?

-in Section 2.2.2, the IR description might be correct, but how is explained the presence of C-Br, C-F and S-CN groups in the IR spectrum? These groups are extremely improbable to be present (of course, C-N, C-O, and -OH are obvious). The comments on IR spectrum should be re-checked, adding supplementary information, as they doesn't seem to be true. Another argument about this is the absence of the elements Cl, Br and S in the EDX spectrum from section 2.2.4.

-in Figure 8, about degradation of MB, there is a specie denoted as .OH-, like an anion radical of hydroxyl, this one is not real, should be a mistake.

-in the Experimental part, the synthesis of AgNP was performed on 16.9 mg of metal precursor, while the IR was dome on 300 mg of Ag NP, is this true?

The work contains some interesting data, but these should be presented in a more specialized way, avoiding routine. More improvement in English and scientific procedures should be done also. Besides, there very little explanation about bringing together the biological activity of Ag NP together with catalytic one, this needs a dedicated correlation. 

Based on these remarks, the manuscript cannot be published in this form, but can be improved, after carefully addressing all comments.

Author Response

(The authors gave the same response as above.)

Round 2

Reviewer 2 Report

The revised manuscript has some major improvements, however the style/English seems to need more attention. Please address this before publication.